# Low threshold and efficient multiple exciton generation in halide perovskite nanocrystals

Mingjie Li[1], Raihana Begum[2], Jianhui Fu[1], Qiang Xu [1], Teck Ming Koh[2], Sjoerd A. Veldhuis[2], Michael Grätzel[3], Nripan Mathews[2,4], Subodh Mhaisalkar [2,4] & Tze Chien Sum [1]

Multiple exciton generation (MEG) or carrier multiplication, a process that spawns two or more electron–hole pairs from an absorbed high-energy photon (larger than two times bandgap energy $E_g$), is a promising way to augment the photocurrent and overcome the Shockley–Queisser limit. Conventional semiconductor nanocrystals, the forerunners, face severe challenges from fast hot-carrier cooling. Perovskite nanocrystals possess an intrinsic phonon bottleneck that prolongs slow hot-carrier cooling, transcending these limitations. Herein, we demonstrate enhanced MEG with $2.25E_g$ threshold and 75% slope efficiency in intermediate-confined colloidal formamidinium lead iodide nanocrystals, surpassing those in strongly confined lead sulfide or lead selenide incumbents. Efficient MEG occurs via inverse Auger process within 90 fs, afforded by the slow cooling of energetic hot carriers. These nanocrystals circumvent the conundrum over enhanced Coulombic coupling and reduced density of states in strongly confined nanocrystals. These insights may lead to the realization of next generation of solar cells and efficient optoelectronic devices.

[1] School of Physical and Mathematical Sciences, Nanyang Technological University, 21 Nanyang Link, 637371 Singapore, Singapore. [2] Energy Research Institute @ NTU (ERI@N), 50 Nanyang Drive, Research Techno Plaza, X-Frontier Block, Level 5, 637553 Singapore, Singapore. [3] Laboratory of Photonics and Interfaces, Department of Chemistry and Chemical Engineering, Swiss Federal Institute of Technology, Station 6, CH-1015 Lausanne, Switzerland. [4] School of Materials Science and Engineering, Nanyang Technological University, 50 Nanyang Avenue, 639798 Singapore, Singapore. These authors contributed equally: Mingjie Li, Raihana Begum. Correspondence and requests for materials should be addressed to S.M. (email: Subodh@ntu.edu.sg) or to T.C.S. (email: Tzechien@ntu.edu.sg)

Hot carriers are formed when a material absorbs photons with energies larger than its bandgap ($E_g$). One of the major limiting factors in photovoltaics is the excess energy loss of the hot carrier via phonon emission. Creating multiple electron–hole pairs per photon can mitigate these energy losses and boost the solar collection efficiencies beyond the Shockley–Queisser (SQ) limit[1,2]. In bulk semiconductors, carrier multiplication (CM) (also termed multiple exciton generation (MEG) in nanocrystals) by impact ionization has been well-documented over the past 3 decades[3]. However, the low-multiplication efficiencies coupled with higher relative threshold photon energies ($h\nu$ larger than $5E_g$) render this effect inconsequential for bulk semiconductor solar cells, owning to fast phonon emission and combined requirements of energy and momentum conservation[4,5]. Subsequently, semiconductor nanostructures, especially strongly confined nanocrystals (NCs)[6–15], were projected to exhibit enhanced MEG due to relaxed momentum conservation[16], enhanced Coulomb coupling[17] and prolonged hot carrier cooling arising from the predicted phonon bottleneck effect under strong quantum confinement[18]. Nevertheless, the phonon bottleneck effect in conventional semiconductors NCs failed to materialize (i.e., faster hot-carrier cooling in smaller NCs)[19–21]. Moreover, the strongly confined NCs also required even higher MEG threshold photon energies than those needed for bulk semiconductors[8,22,23], which is attributed to the small density of final states in quantum confined system that cancelled the volume confinement effects on the Coulomb coupling matrix elements[22]. Therefore, MEG candidates with both enhanced MEG efficiency and low threshold photon energy are highly desired.

Intrinsic phonon bottleneck leading to long hot-carrier lifetimes have recently been reported in lead bromide perovskite NCs[24]. Since the MEG process competes with hot-carrier cooling, slow hot-carrier cooling observed in perovskite NCs renders this material a highly promising system for MEG. In this work, we report that intermediate-confined formamidinium lead iodide (FAPbI$_3$) NCs, possessing an intrinsic phonon bottleneck, transcend the above challenges as an emerging class of NCs for enhanced MEG. We demonstrate enhanced MEG with threshold down to $2.25E_g$ and slope efficiency up to 75% in intermediate-confined colloidal FAPbI$_3$ NCs, surpassing the reported MEG slope efficiencies in strongly confined incumbents, viz., lead sulfide (PbS) or lead selenide (PbSe)[25]. Efficient MEG occurs via inverse Auger process within 90 fs, afforded by the slow cooling of energetic hot carriers. These perovskite NCs circumvent the conundrum over the increased carrier cooling rates and reduced density of states, which reversed the gains of enhanced Coulomb coupling and relaxation of momentum conservation in strongly confined conventional NCs. These insights may lead to the realization of a next generation of solar cells and efficient light-harvesting devices.

## Results

**Synthesis and characterization of FAPbI$_3$ NCs.** Cubic-shaped colloidal FAPbI$_3$ NCs prepared through hot-injection synthesis were size-segregated (7.5–13 nm edge length) using different centrifugation speeds and ligands (see Supplementary methods and TEM images in Supplementary Fig. 1). X-ray diffraction (XRD) characterization validates the formation of the perovskite α-phase (Supplementary Fig. 2). Obvious blue-shift of PL maxima from 805 to 750 nm were observed for NCs of reduced size (Fig. 1a) along with the increased $E_g$ from 1.56 to 1.7 eV (Fig. 1b, see Supplementary Fig. 3 and Supplementary Note 1 for details of determination of $E_g$). From effective-mass approximation (EMA), the exciton Bohr diameter of FAPbI$_3$ is $12.3 \pm 0.2$ nm

(Supplementary Note 1), our NC samples are thus in the intermediate to weak confinement regime. A solution processed polycrystalline FAPbI$_3$ film with $E_g = 1.54$ eV ($750 \pm 10$ nm in thickness, see Supplementary Fig. 4) and colloidal PbS NCs (around 3.5 nm in diameter, $E_g = 1.3$ eV) were also investigated for comparison as the bulk-counterpart and the reference of strongly confined conventional NCs, respectively.

Transient absorption (TA) spectroscopy was performed on FAPbI$_3$ NCs solution in 1-mm quartz cell under continuous stirring to prevent potential distortion of TA signals due to photocharging[25,26]. The biexciton Auger recombination (AR) lifetime ($\tau_{xx}$) of NCs was determined using pump fluence-dependent TA dynamics with $h\nu$ smaller than $2E_g$ (Supplementary Figs. 5–7). For NCs size below the Bohr exciton volume (vertical blue dashed line in Fig. 1c), $\tau_{xx}$ follows a volume dependence with $\tau_{xx}$ proportional to $V^{0.7}$, which is slightly steeper than those reported for other weakly confined perovskite NCs ($\tau_{xx}$ proportional to $V^{0.5}$)[24,27], and less than the linear dependence for conventional strongly confined semiconductor NCs[28]. Furthermore, the measured $\tau_{xx}$ of $400 \pm 40$ ps for our largest NCs with size of 12.9 nm is consistent with the reported value (red square in Fig. 1c) in FAPbI$_3$ NCs with size of 13 nm[29].

**Determination of MEG in FAPbI$_3$ NCs.** The distinct TA dynamics of multiple excitons with lifetime usually ranging from tens to hundreds of ps at low-pump fluence and high pump energies (larger than $2E_g$) (compared to several to tens of ns for single-exciton lifetimes) provides validation of the MEG signature. For pump photon energies below $2E_g$ (energy-conservation limit for MEG) at low-pump intensity ($<N_0>$ smaller than around 0.3), the band-edge photobleaching (PB) dynamics exhibit a single exponential decay (Fig. 2a, b). This not only indicates neutral single-exciton decay, but also the absence of any appreciable population decay that may originate from trions (i.e., charged excitons). With increasing pump photon energy $h\nu$ above $2E_g$, an obvious fast decay component with increased amplitude emerges (Fig. 2a). Such decay is consistent with the biexciton lifetime. Notably, such fast biexciton decay with lifetime of $160 \pm 10$ ps for 7.5 nm-sized NCs (Fig. 2c and Supplementary Fig. 5) persists at the extremely low-pump intensity of $<N_0> = 0.07$ at $h\nu = 2.7E_g$ (corresponding to pump fluence of $0.12 \mu J \, cm^{-2} \, s^{-1}$ photon$^{-1}$ and $1.63 \times 10^{11}$ photons cm$^{-2}$ s$^{-1}$; and in fact, this low fluence is comparable to that under 1-sun irradiance at similar photon energies). Similar phenomena were also observed in larger-sized FAPbI$_3$ NCs (Supplementary Figs. 6 and 7), but absent in the bulk-counterpart (Fig. 2a inset). This appearance of fast biexciton decay at low-pump fluence and high-pump phonon energy provides the initial evidence for MEG.

To quantify MEG, its quantum yield (QY; i.e., the average number of excitons generated per absorbed photon) is determined. In the presence of MEG, the initial peak PB is proportional to $<N_0> \cdot$ QY. When the delay time after photoexcitation is much longer than multiexciton recombination, each NC will recombine through single-exciton decay. Hence, at longer delays, the PB signal is proportional to the population of photoexcited NCs in the photoexcited volume, which can be described by $1 - \exp(-<N_0>)$ according to Poisson statistics. Thus, the exciton population ratio ($R_{POP}$) of excitons generated directly following excitation and at long delays when AR is complete (herein, at $\Delta t$ of 4 ns) can be described by[9]:

$$R_{POP} = \frac{\delta <N_0> \cdot QY}{1 - \exp(-<N_0>)} \quad (1)$$

where $\delta$ refers to single-exciton decay. From pump fluence-dependent PB dynamics, $R_{POP}$ vs. $J_0$ or $<N_0>$ can be obtained

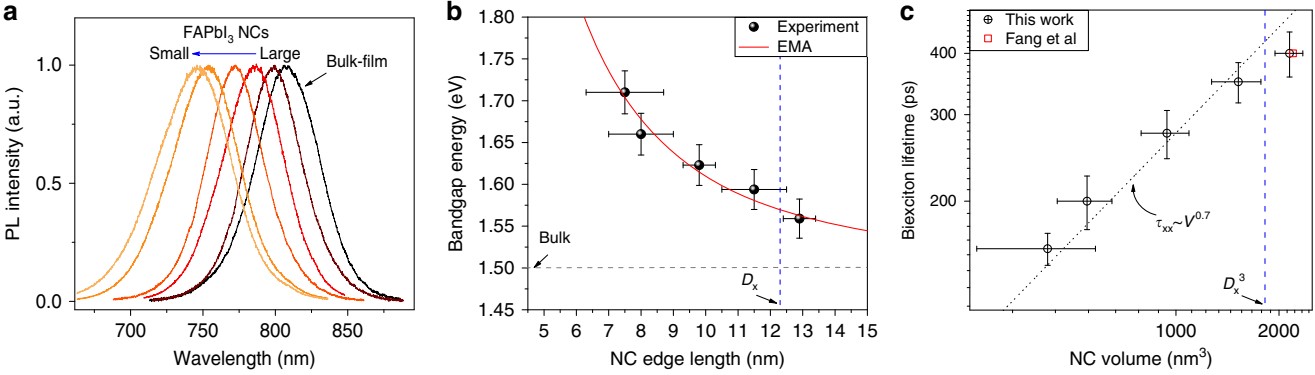

**Fig. 1** Size-dependent bandgaps and biexciton lifetimes of FAPbI$_3$ NCs. **a** Normalized PL spectra of FAPbI$_3$ NCs in toluene with different sizes (emission peak positions are listed in Supplementary Table 1) and the FAPbI$_3$ polycrystalline bulk-film counterpart. **b** Bandgap energies (black dots) as a function of the edge length of FAPbI$_3$ NCs together with fitting using Supplementary Eq. (1) (red solid line). The vertical blue dashed line indicates the position of exciton Bohr diameter ($D_x$). Error bars correspond to the NC size distribution and uncertainty in the measurements of TA spectra. **c** Biexciton Auger-lifetime (black circles) as a function of NC volume. The horizontal and vertical error bars correspond to the NC size distribution and the uncertainty in the fitting procedure, respectively. The red square is obtained from ref. [29] (Fang et al)

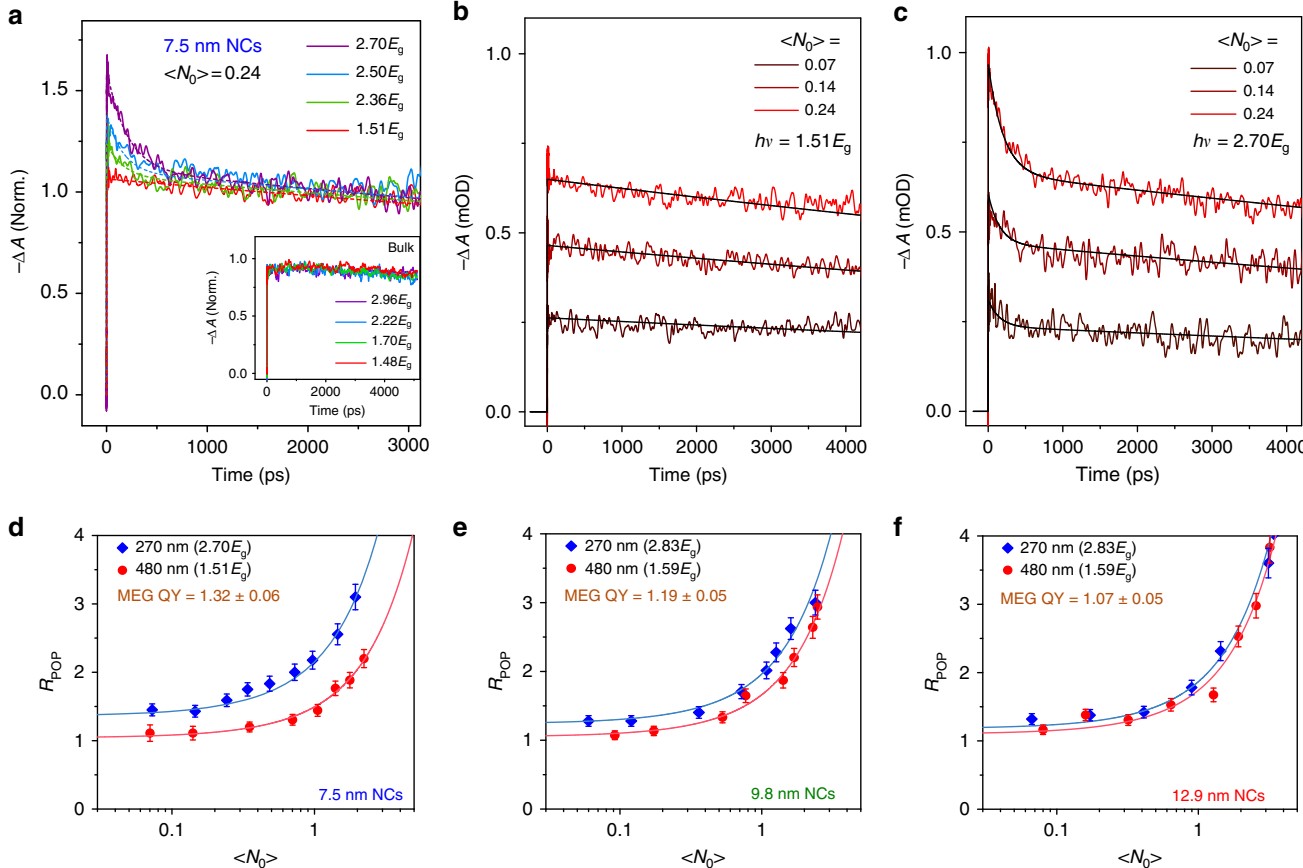

**Fig. 2** MEG QY determination from TA dynamics. **a** Normalized (norm.) band-edge PB dynamics under different pump photon energies with $\langle N_0 \rangle = 0.24$ in 7.5 nm-sized FAPbI$_3$ NCs. Inset shows the similar measurements for the bulk-counterpart with pump fluence of $2 \times 10^{17}$ cm$^{-3}$. TA dynamics generated by photon energy of (**b**) $1.51E_g$ and (**c**) $2.70E_g$ for 7.5-nm sized NCs with different average number of absorbed photons per NC $\langle N_0 \rangle$. Solid black lines are single exponential in (**b**) and biexponential fittings in (**c**), respectively. **d–f** $R_{POP}$ as a function of $\langle N_0 \rangle$ under photon energies below and above MEG threshold for three different sized FAPbI$_3$ NCs. $R_{POP}$ is determined by the PB intensity ratio at the beginning of pump excitation at a delay time of 2 ps to that at a delay time of 4 ns and fitted with Eq. (1) (solid lines). The error bars in **d–f** represent the uncertainties in the determination of TA amplitudes in the fitting procedure of TA dynamics

(Fig. 2d–f and Supplementary Figs. 5–7). The accuracy in determining $\langle N_0 \rangle$ is ensured by cross-checking the absorption cross-sections (Supplementary Note 2). Firstly, $R_{POP}$ is larger at higher photon energies, indicating that more excitons are generated at the same initial $\langle N_0 \rangle$. Secondly, at very low fluence ($\langle N_0 \rangle$ smaller than 0.2) for both low- and high-pump photon energies, $R_{POP}$ reaches a near fluence independent regime, indicating the absence of multiphoton absorption, and ensuring the accurate measurement of MEG QY[30]. MEG QY is determined by the ratio of the $y$-intercepts of fitting curves of $R_{POP}$ (using Eq. (1)) with $\langle N_0 \rangle$ much smaller than 0.1 at above and below MEG threshold (Fig. 2d–f). For 7.5 nm-sized NCs, the MEG QY is 1.32 ± 0.06 at $h\nu = 2.7E_g$. For larger NCs, MEG QY reduces to 1.19 ± 0.05 at $2.83E_g$ and 1.07 ± 0.05 at $2.94E_g$ for 9.8 and 12.9 nm-sized NCs, respectively. It is worth noting that, MEG was not observed in an earlier report on weakly confined CsPbI3 NCs (with 11.2 nm edge length, $E_g$ 1.82 eV, and exciton Bohr diameter of around 12 nm) at $2.65E_g$ excitation[27].

**Enhanced MEG in smaller perovskite NCs.** Figure 3a shows the MEG QY as a function of pump photon energy relative to $E_g$ ($h\nu_{th}/E_g$). The threshold for the onset of MEG is obvious for the smallest NCs, which reduces drastically for the larger NC sizes. For 7.5 nm-sized NCs, the MEG threshold is as low as $2.25E_g$ (or $2.3 ± 0.1E_g$), which is smaller than the approximately $3E_g$ for conventional strongly confined PbS and PbSe NCs (i.e., the benchmark materials in the MEG NCs discussion, see Fig. 3b, Supplementary Fig. 8 and Supplementary Table 2). Moreover, both $h\nu_{th}/E_g$ and the actual $h\nu_{th}$ values of our intermediate-confined perovskites NCs follow a similar trend (i.e., smaller threshold at smaller NC sizes—Fig. 3a and Supplementary Fig. 9). In contrast, this is reversed for strongly confined conventional NCs. Despite their smaller $h\nu_{th}/E_g$ values due to the larger $E_g$[8,23], the actual $h\nu_{th}$ values increase for conventional NC of smaller sizes. In addition, the reduced final density of states in strongly confined NCs will further increase $h\nu_{th}$ and reduce MEQ QY[26]. These results suggest that intermediate confinement is more beneficial for efficient MEG.

Apart from the MEG threshold and MEG QY, another important MEG property is the slope efficiency ($\eta_{MEG}$), which represents the increment of additional excitons with respect to the increased photon energy beyond the MEG threshold. The $\eta_{MEG}$ can be obtained from the slope of the MEG QY vs. $h\nu_{th}/E_g$ plot. For our perovskite NCs, $\eta_{MEG}$ is determined to be 75%, 50% for 7.5 and 9.8 nm-sized NCs, respectively (Fig. 3a). In comparison, PbS and PbSe NCs exhibit $\eta_{MEG}$ values of around 40% (Fig. 3b

and Supplementary Table 2). Furthermore, $\eta_{MEG}$ and $h\nu_{th}$ follow a simple empirical relation[7]: $h\nu_{th} = \left(1 + 1/\eta_{MEG}\right)E_g = E_g + \varepsilon_{eh}$, where $\varepsilon_{eh}$ represents the energy increment of an incident photon required to produce a new exciton at energies above $h\nu_{th}$. From our experimentally determined $\eta_{MEG}$, we calculated $\varepsilon_{eh}$ of ca. $1.33E_g$ and $2E_g$, for 7.5 and 9.8 nm-sized NCs, respectively, which is close to the experimentally observed values (i.e., $1.25E_g$ and $1.5E_g$, respectively).

The lower-MEG threshold in intermediate-confined perovskite NCs affords significant advantages for enhancing the solar power conversion efficiency (PCE). Figure 3c shows the calculated PCE based on the detailed balance equation (see details in Supplementary Note 3), assuming 100% MEG efficiency under AM1.5 (1 sun) illumination for different material bandgaps. The peak PCE can be significantly improved compared to the SQ limit (ca. 33%) to be larger than 40% when $h\nu_{th}$ is close to $2E_g$. However, at $h\nu_{th}$ larger than $3E_g$ MEG has negligible effect on the overall efficiency improvement. The PCE at different $\eta_{MEG}$ is also calculated (Supplementary Fig. 10) as reference. The low-MEG threshold approaching the energy conservation limit in perovskites NCs would therefore be highly favorable for improving the efficiency of PVs.

**Slow cooling of high energy hot carriers for efficient MEG.** We next examine the cooling dynamics of highly excited hot carriers to understand the mechanism underlying a more efficient MEG in the smaller FAPbI3 NCs. Although several mechanisms (i.e., impact ionization[31], direct multiexciton generation via a virtual exciton state[32], and coherent superposition of single and multiple exciton states[33]) were proposed to illustrate the appearance of MEG in conventional NCs, none have been verified yet. Analysis of the dynamical signatures will thus divulge much insight into the underlying mechanism. Figure 4a (right panel) shows a schematic of the MEG process initiated by hot carriers photo-excited with $h\nu$ larger than $2E_g$ via the inverse of AR. The rate for the MEG process ($1/\tau_{MEG}$) must be higher than the cooling rate for the hot carriers ($1/\tau_{cool-int}$), where $\tau_{MEG}$ refers to the MEG lifetime and $\tau_{cool-int}$ is the interval cooling time from the initial photoexcited state to the state constituting the MEG threshold, (i.e., $\tau_{MEG} < \tau_{cool-int}$). The total cooling time of hot carriers from various photoexcited states to the band-edge ($\tau_{cool}$) can be obtained from the build-up (or rise) times of the band-edge PB signature (Fig. 4b). The interval cooling time $\tau_{cool-int}$ can thus be obtained from $\tau_{cool}$ (at $h\nu > 2E_g$) − $\tau_{cool}$ (at $h\nu = 2E_g$). To avoid the convolved effects from the hot-phonon bottleneck (at around $10^{18}\,cm^{-3}$) and longer hot-carrier cooling at higher carrier

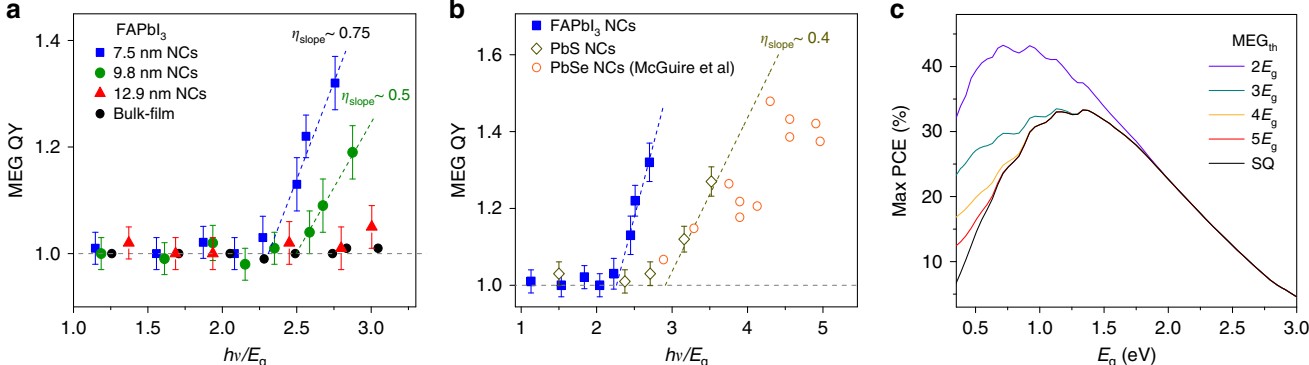

**Fig. 3** MEG threshold and slope efficiency determination. **a** MEG QY as a function of relative pump photon energies ($h\nu/E_g$) for FAPbI3 NCs of different edge lengths and bulk-counterpart. **b** Comparison of MEG QY vs. $h\nu/E_g$ with PbSe NCs (from ref. [25]) and PbS NCs ($E_g = 1.3$ eV) as a reference measured using the same setup. The error bars in **a**, **b** represent the uncertainties in the MEG QY fitting procedure. **c** Detailed balance calculations for maximum PCEs under AM1.5 solar illumination as a function of material $E_g$ for different MEG thresholds. The Shockley–Queisser limit is denoted as SQ

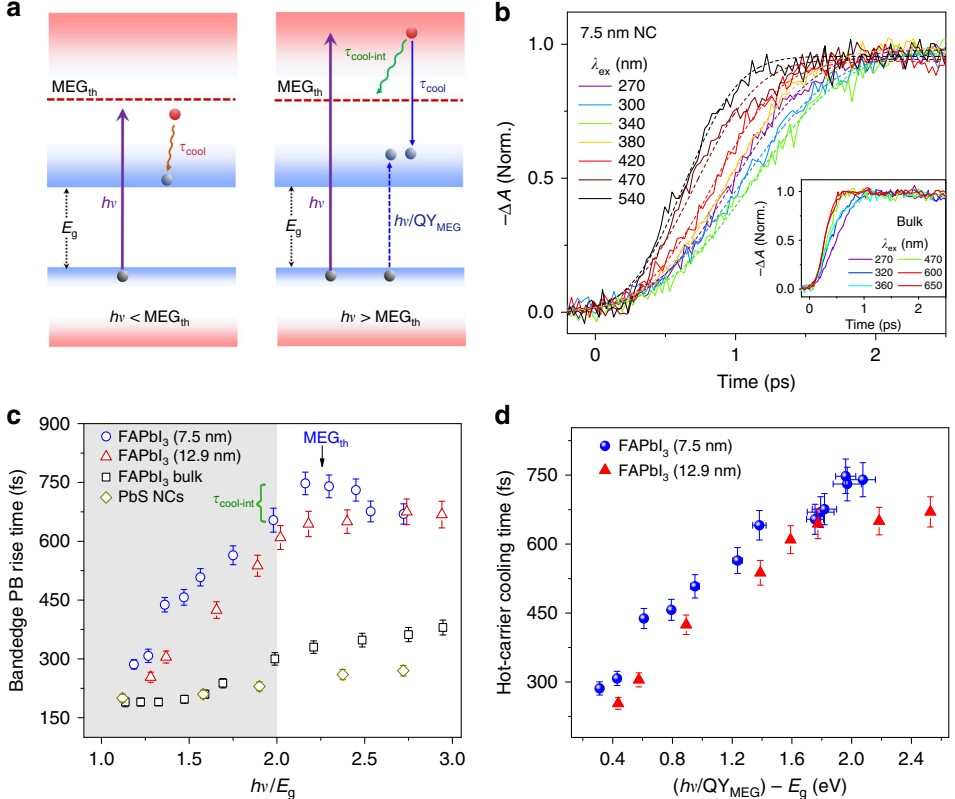

**Fig. 4** Slower hot-carrier cooling in smaller FAPbI$_3$ NCs. **a** Schematic for hot-carrier cooling (left) below MEG threshold and (right) above the MEG threshold. **b** Normalized band-edge PB dynamics in colloidal 7.5 nm-sized FAPbI$_3$ NCs and their bulk-counterpart (inset) under different pump wavelengths. The dashed lines are single exponential fits. **c** Band-edge PB buildup-time of FAPbI$_3$ NCs (7.5 nm (blue circles); 12.9 nm (red triangles)), polycrystalline bulk-film (black square), and PbS NCs (yellowish-brown diamonds) measured at different pump photon energies. Gray region indicates the energies below ideal MEG threshold of 2$E_g$. **d** Hot-carrier cooling time as a function of converted real excess energies of carriers in FAPbI$_3$ NCs. The error bars in (**c**) and (**d**) represent the uncertainty in the fitting procedure for the rise time

densites[24], the effective volume carrier density in NC and bulk samples were kept constant at $2 \times 10^{17}$ cm$^{-3}$.

For FAPbI$_3$ bulk-film (black squares), $\tau_{cool}$ gradually increases from 160 fs (comparable with the instrument response) to 375 fs at 3$E_g$ photoexcitation (Fig. 4b inset and Fig. 4c). These values are consistent with previous reports for polycrystalline halide perovskite films (i.e., MAPbI$_3$[34] and MAPbBr$_3$[24], MA = methylammonium) at the similar carrier densities, which is due to the strong carrier-phonon coupling in the polar perovskites. Surprisingly, $\tau_{cool}$ in FAPbI$_3$ NCs are approximately two times longer than their bulk-counterpart and reference PbS NCs (Fig. 4c and Supplementary Fig. 11). Furthermore, $\tau_{cool}$ is slightly slower in the smaller 7.5 nm-sized NCs below 2$E_g$ threshold, which is consistent with our previous observation of the intrinsic photon-bottleneck in MAPbBr$_3$ NCs[24]. Nevertheless, it should be noted that highly excited hot carriers above the 2$E_g$ threshold (Fig. 4a, red dashed line) relax faster because the energy states are more closely-spaced and the electronic structure is reminiscent of those in bulk materials. Indeed, for large sized FAPbI$_3$ NCs, $\tau_{cool}$ exhibits an initial fast increase that slows down considerably above 2$E_g$ (Fig. 4c). The increased competition with hot-carrier cooling thus considerably hinders the occurrence of MEG.

Interestingly, the total hot-carrier cooling times become slightly shorter at higher pump photon energies beyond MEG threshold in the smaller 7.5 nm-sized NCs (Fig. 4c). This can be explained that MEG occurs via inverse Auger process (Fig. 4a) will result in shorter hot-carrier cooling times (Supplementary Fig. 12). After such MEG process, the generated multiple excitons will share the

"absolute" excess energy of that particular pump photon ($h\nu - E_g$). Thus, the hot-carrier cooling lifetime to the band-edges is approximately proportional to the excess energy produced by each exciton: $\tau_{hot}$ proportional to ($h\nu/QY_{MEG} - E_g$). Consistently, the observed hot-carrier cooling lifetimes monotonously increase with ($h\nu/QY_{MEG} - E_g$) (Fig. 4d). These results, therefore, validate that the MEG mechanism in our NCs is based on the inverse Auger process[31,35].

Lastly, $\tau_{cool-int}$ for our 7.5 nm-sized NCs is determined to be 90 $\pm$ 5 fs (Fig. 4c—green bracket), indicating that $\tau_{MEG}$ is smaller than 90 fs. In contrast, much shorter interval cooling time are observed for other samples in Fig. 4c (around 30 fs)—evident from the gradual increase of the hot-carrier cooling lifetimes at higher $h\nu/E_g$ values; consistent with their higher MEG threshold. In addition, $\tau_{MEG}$ will also be faster in smaller NCs given the enhanced carrier Coulombic coupling (i.e., reflected by the enhanced AR in smaller perovskite NCs—Fig. 1c). This allows MEG to compete more effectively with the fast hot-carrier cooling at higher energy states in our 7.5 nm-sized NCs.

## Discussion

From our density function theory (DFT) calculations (Supplementary Fig. 13), the electron and hole effective masses are found to be 0.14$m_0$ and 0.20$m_0$, respectively. The small and similar effective masses of the electron and hole are predicted to be favorable for slow hot-carrier cooling and more efficient MEG[36,37]. Furthermore, another recent computational work[38] also demonstrated low-threshold MEG (close to 2$E_g$) in MAPbI$_3$ NCs. The authors attributed that the stronger Coulomb coupling

between the initial single-exciton and final-biexciton states, and the longer hot-carrier cooling of highly excited states were more conducive for MEG. Their calculated MEG process was on a time scale of tens of fs, which agrees well with our measured value of smaller than 90 fs. Although, PbSe or PbS also have similar small carrier effective masses, the high-MEG threshold can be explained by their faster hot-carrier cooling (Fig. 4c) and/or reduced density of final states in strongly confined NCs. We, therefore, ascribe the origins of the low threshold and efficient MEG in our intermediate-confined perovskite NCs to arise from the small and similar carrier effective masses, slow hot-carrier cooling and appropriate quantum confinement effect.

In summary, we observed enhanced MEG efficiencies (up to 75%) and low-MEG threshold energies (down to $2.25E_g$) in intermediate-confined $FAPbI_3$ NCs. The low-MEG threshold provides a distinct advantage for PCE enhancement. Our results further reveal that the intermediate-confined perovskite NCs overcome the fast hot-carrier cooling limit, and bypass the conundrum over the contrasting effects between enhancing the Coulombic coupling and the reduced density of states in strongly confined NCs for MEG. Engineering perovskite NCs with even smaller bandgaps (e.g., with Sn-doping) could lead to even better performance in photovoltaic applications. These insights may therefore demonstrate a direction for developing optoelectronic concepts towards highly efficient solar energy conversion devices.

## Methods

**Synthesis of $FAPbI_3$ NCs**. Synthesis of formamidinium oleate (FA-oleate): formamidinium acetate (0.26 g, 2.5 mmol, Sigma-Aldrich, 99%) along with 5 mL of dried oleic acid (Sigma-Aldrich, 90%) was heated and stirred under vacuum condition at 120 °C for ca. 30 min. Upon complete dissolution of formamidinium acetate, vacuum was stopped and the mixture purged with $N_2$ gas for several min. The FA-oleate was preheated to 100 °C before injection.

Synthesis of formamidinium propionate (FA-propionate): For this synthesis, 5 mL of propionic acid (Sigma-Aldrich, 99.5%) was added to 0.26 g of formamidinium acetate. First, under vacuum at room temperature for 15 min, followed by heating at 80 °C for 15 min under $N_2$. The solution was cooled to room temperature before injection.

Synthesis of $FAPbI_3$ NCs: $PbI_2$ (0.086 g, Sigma-Aldrich, 99%) was taken along with 5 mL of dried 1-octadecene (ODE) in a two-neck round bottom flask. To this, 1 mL of dried oleic acid and 650 μL of dried oleylamine were injected under vacuum, followed by heating at 120 °C for approximately 45 min. After complete dissolution of $PbI_2$, the temperature was reduced to 80 °C after which 1 mL of FA-oleate (preheated) or FA-propionate (kept at room temperature) was injected to achieve the desired size (see Supplementary Table 1, OA = oleate, PA = propionate). After 10 s, the NCs were cooled by an ice-water bath.

Purification and centrifugation: The crude NCs in ODE were centrifuged at different centrifugal speed to obtain the desired size. The batches prepared with FA-propionate were first centrifuged at 1000 r.p.m., the precipitate was collected to get bigger NCs with average edge length of 12.9 nm, while the supernatant in ODE was centrifuged once more at 5000 r.p.m., after which the precipitate was collected to get smaller NCs with average edge length of 11.5 nm. The above two precipitates were redispersed in 5 mL of anhydrous toluene (Sigma-Aldrich), vortexed and centrifuged again at 3000 r.p.m. to discard bigger NCs or any other agglomerated particles.

Similarly, the batches prepared with FA-oleate were centrifuged at three different centrifugal speeds: 3000–7000 r.p.m., and 11,000 r.p.m. Each time, the precipitate was collected, and the supernatant phase in ODE was subsequently centrifuged at higher centrifuge speeds. Finally, the supernatant phase obtained after centrifugation at 11,000 r.p.m. was discarded. The obtained precipitates (obtained at three different speeds) were redispersed in anhydrous toluene, vortexed and centrifuged once more at 3000 r.p.m. to achieve homogenous NCs solutions with average edge lengths of around 9.8, 8.0, and 7.5 nm, respectively, which were further used for various characterizations and spectroscopic measurements. The as-synthesized NCs were stable for more than three weeks both in solution and film in a glovebox under nitrogen atmosphere.

**$FAPbI_3$ film preparation**. $PbI_2$ and FAI were purchased from TCI and Dyesol, respectively. $FAPbI_3$ perovskite solution was prepared by mixing 461 mg of $PbI_2$ and 172 mg of FAI in DMF solvent and stirring it for 1 h. $FAPbI_3$ perovskite film was prepared by spin coating 1 mol $dm^{-3}$ perovskite solution at 4000 r.p.m. for 12 s while chlorobenzene (antisolvent) was dropped at 9 s.

**PbS NCs**. PbS NCs (diameter around 3.5 nm) in toluene were purchased from Evident Technologies.

**Characterizations**. Powder XRD patterns were recorded using a XRD Bruker D8 Advance with Cu Kα radiation ($\lambda = 1.5418$ Å). The sizes of the NCs were determined by using a Tecnai transmission electron microscope (TEM, model FEI Tecnai F20), samples were dropcasted on carbon-coated copper grid. SEM images of the perovskite film were taken by a field emission scanning electron microscope (FESEM) JEOL JSM-7600F.

**TA spectroscopy**. TA spectroscopy (with pump beam diameter of 0.5 mm and probe beam diameter of 0.3 mm) was performed using a Helios$^{TM}$ setup (Ultrafast Systems LLC) with chirp-correction. The pump pulses were generated from a Coherent TOPAS-C optical parametric amplifier pumped by a 1 kHz regenerative amplifier (Coherent Legend, 800 nm, 150 fs). The amplifier was seeded by a mode-locked Ti-sapphire oscillator (Coherent Vitesse, 100 fs, 80 MHz). The white light continuum probe beam (in the range from 400 to 850 nm) was generated from a 3-mm sapphire crystal using 800-nm pulse from the regenerative amplifier mentioned above. The probe beam passing through the sample was collected using a detector for UV–Vis (CMOS sensor). All measurements were performed at room temperature with NCs samples in toluene in a 1 mm-thick quartz cell and stirred with a magnetic stirrer, the film sample is in a $N_2$-filled quartz chamber during measurements. The pumping intensity in NCs was controlled by the average number of absorbed photons per NC ($<N_0>$), where $<N_0> = J_0\sigma_P$, $\sigma_P$ is the absorption cross-section at pump wavelength and $J_0$ is pump fluence.

**DFT calculations**. The electronic structures of cubic $FAPbI_3$ with spin-orbital coupling were calculated using the all electron-like projector augmented wave method[39], and the Perdew–Burke–Ernserhof (PBE) exchange correlation potential[40] as implemented in the VASP code[41]. The semicore of Pb atoms ($5d$ orbital) was treated as valence electrons, i.e., 14 valence electrons for Pb ($5d^{10}6s^26p2$) atom. The cut-off energy for the plane wave expansion of the wave functions was 500 eV. The primitive cell was completely optimized, including its lattice vectors and atomic positions. The Hellman–Feynman forces were less than of 1 meV $Å^{-1}$. The $6 \times 6 \times 6$ Monkhorst-Pack grid of k-points for Brillouin zone integration of $FAPbI_3$ were used in the optimization of crystal structure.

## Data availability

The data supporting the plots within this paper and other findings of this study are available from the corresponding authors upon request.

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

## Acknowledgments

Financial support from Nanyang Technological University start-up grant M4080514; JSPS-NTU Joint Research Project M4082176; the Ministry of Education AcRF Tier 1 grant RG173/16 and Tier 2 grants MOE2015-T2-2-015 and MOE2016-T2-1-034; and from the Singapore National Research Foundation through the Competitive Research Program NRF-CRP14-2014-03 and the NRF Investigatorship Program NRF-NRFI2018-04 are gratefully acknowledged. Funding from Office of Naval Research Global (ONRG-NICOP-N62909-17-1-2155) is also acknowledged.

## Author contributions

T.C.S., M.L., N.M., and S.M. conceived the idea and designed the experiments. M.L. performed the spectroscopic characterization. R.B., T.M.K., and S.A.V. prepared the NCs and film samples and performed the sample characterization. J.F. and Q. X. performed the PCE and DFT calculations, respectively. M.L., M.G., N.M., S.M., and T.C.S. analysed the data and wrote the manuscript. All authors discussed the results and commented on the manuscript at all stages. T.C.S. and S.M. led the project.

## Additional information

**Competing interests:** The authors declare no competing interests.

