## [Peer Review File · Nature Communications]

Reviewers' comments:

Reviewer #1 (Remarks to the Author):

This interesting paper by Li et al reports on the MEG efficiencies in FAPbI₃ nanocrystals with various sizes and degrees of quantum confinement. The authors demonstrate that intermediate confined nanocrystals can represent a good balance to overcome the hot-carrier limit and the reduced density of states present in too strongly quantum confined nanocrystals.

The intrinsic phonon bottleneck of perovskite nanocrystals, that was previously reported by the authors (Nat Comms, 8, 2017), can be one of the causes favoring the appearance of MEG in such systems.

I think that the results are well argued and are definitely valuable for the readership of Nature Communications. I therefore strongly recommend the acceptance of this work in Nature Communications.

However, there are a few important aspects that should be addressed before publishing:

1. The authors always refer their results to multiples of the bandgap energy E_g . In the case of quantum confined nanocrystals, it would be more appropriate to refer to the exciton energy instead. If for simplicity reasons you want to keep the E_g nomenclature, you should at least comment on the exciton binding energy of these perovskite nanocrystals and, if this is non negligible, relate these results to multiples of the exciton energy.
2. Calculating the efficiency of the MEG process, the authors should clarify whether they refer to 2-exciton generation only or if they consider also possible higher order MEG (generating a number of excitons ≥ 3). These high-order processes should be taken into account, as they use as excitation energies also values that are higher than 3 times the bandgap energy.
3. In the balance calculation reported in Fig. 3c, do you exclude the generation of 3 or more excitons (when allowed by the energy conservation)?
4. The authors should clarify in the text the method they have used to calculate the MEG QY.

Reviewer #2 (Remarks to the Author):

In the present manuscript, the authors explore the implications of their recent work (M.J. Li et al., Nat. Comm. 8, 14350 (2017)) showing a phonon bottleneck in the cooling of hot carriers in lead halide perovskite nanocrystals (NCs). The slowing of hot carrier cooling observed in their earlier work suggests that the efficiency of carrier multiplication (CM) should increase in lead halide perovskite nanocrystals compared to the bulk material. The observation of slower cooling with increasing confinement in their earlier report opens the possibility that the efficiency of CM could be higher in lead halide perovskites than in traditional inorganic NCs like PbS and PbSe. The present manuscript illustrates that these expectations are born out. Specifically, the threshold for the onset of CM in FAPbI₃ NCs is found to be as low as 2.25 ± 0.1 times the gap in 7.5 nm diameter NCs, while the slope efficiencies are observed to be as high as 75% (i.e., an average of 0.75 extra electron-hole pairs are

generated per additional gap of photon energy above the CM threshold). These numbers are substantially better than in the most widely studied systems, PbS and PbSe, which had once been thought to offer the greatest potential for CM. Importantly, the efficiencies reported here are most favorable in the smallest dots, which suggests room for further improvement. This is again in contrast to the case of traditional inorganic NCs in which carrier cooling rates increase with increasing confinement, thereby negating the gains in Coulomb interactions and relaxation of momentum conservation that come with increased confinement and that would otherwise be expected to lead to increased CM efficiencies.

It should be noted that Klimov's group has shown a CM threshold below 2.2 eV in PbSe/CdSe core/shell NCs (C.M. Cirloganu et al., Nat. Comm. 5, 4148 (2014)), which partly undercuts one of the main claims here, namely that the CM threshold is lower than in traditional inorganic NCs. However, the slope efficiencies were still only 25%. At least as important as the numbers is the fact that such core/shell NCs are fundamentally more complicated than the perovskite NCs studied here.

The study of CM has struggled somewhat since the initial reports of high efficiencies in inorganic NCs were found to be erroneous. CM has been overtaken as a topic of study by singlet fission for a variety of reasons including perceptions of potential efficiencies and the fact that there have been clearer paths to design of molecules for singlet fission. There will be interest in the present work in boosting prospects for CM in a relatively simple material system. As such, this is a valuable contribution to the field. It should also be noted that the data in the present manuscript is very thorough, and great care appears to have been taken to obtain clean data.

The one general shortcoming I see in the paper is the absence of a quantitative explanation for the observations. That the CM efficiencies in FAPbI₃ NCs are greater than in the bulk was to be expected given the authors' previous paper on the phonon bottleneck in lead halide perovskite nanocrystals. In lines 28-30, the authors write that "these NCs solve the conundrum over the contrasting requirements of enhancing Coulombic coupling and the density of states in strongly confined NCs." The results don't really solve the conundrum; they just avoid it by taking advantage of slowed cooling. Like many papers on CM, the paper offers more in the way of materials characterization than new scientific insights. A particularly glaring deficiency is the absence of discussion of the effective masses of electrons and holes. In the conventional picture, for a system characterized by single conduction and valence bands and in the case that the effective masses of electrons and holes are equal, the threshold for CM is $3E_g$. In other lead halide perovskites, the effective masses of electrons and holes are similar to each other (see, for example, Ref. 27 of the present manuscript and L.M. Herz, ACS Energy Letters 2, 1539 (2017)). As noted in lines 152-154 of the present manuscript, the authors of Ref. 27 did not observe MEG in CsPbI₃ under excitation at a photon energy of 2.65 E_g . In fact, it was in terms of the similar electron and hole masses that the failure to observe MEG in Ref. 27 was understood. From this perspective, the low CM threshold reported here is unexpected. It may be that the conventional picture is not relevant here, but such a statement would need to be justified.

The rest of my comments are about specific details of the paper.

Panels c and d in Supplementary Figs. 5 – 7 are somewhat confusing. In particular, what is the physical meaning of using a linear fit for the A data, i.e., for ? With the exception of the 7.5 nm NCs excited below the MEG threshold, the fits are clearly statistically inconsistent with the data insofar as few of the error bars cross the fit line. Moreover, the inconsistencies are in different directions for different data sets. In Figs. 5c and 6d the A data seem roughly linear; in Fig. 5b the data appears to have a sub-linear dependence on fluence; and in Figs. 6c, 7c, and especially 7d, the A data appear to have a super-linear dependence on fluence.

The authors should state that the FA in FAPbI₃ refers to formamidinium the first time FAPbI₃ is mentioned in the manuscript, not just in the methods section.

I presume that the solid red and blue fits to RPOP are fits of Eq. 1. This is implied but not stated explicitly.

Reviewer #3 (Remarks to the Author):

The manuscript "Low threshold and Efficient Multiple Exciton Generation in Halide Perovskite Nanocrystals" describes multiple exciton generation in colloidal FAPbI₃ nanocrystals. The claimed multiple exciton generation (MEG) threshold is very low ($2.25E_g$), being potentially relevant for photovoltaic applications. The positive effect on power conversion efficiency of solar cells is demonstrated by calculations for different MEG thresholds, showing that only the threshold of $2E_g$ is relevant.

The efficiency of the MEG is relatively high due to the phonon bottleneck and slow cooling of hot carriers. The latter has been experimentally studied by the femtosecond transient absorption (TA) measurements, where the band edge bleaching shows a risetime of hundreds of femtoseconds. The dynamics of multiple excitons is studied by TA spectroscopy as well, where the band edge bleaching at pumping photon energies above $2E_g$ exhibits a fast component at the time scale of 100 picoseconds. The conclusions are supported by experiments in a convincing way. The experiment description is detailed enough for reproduction.

- All reviewer comments are displayed in *dark orange italics*.
- All our responses are displayed in black.
- Sentences indicating changes to the manuscript are blue.

Reviewer #1:

General Comments: *This interesting paper by Li et al reports on the MEG efficiencies in FAPbI₃ nanocrystals with various sizes and degrees of quantum confinement. The authors demonstrate that intermediate confined nanocrystals can represent a good balance to overcome the hot-carrier limit and the reduced density of states present in too strongly quantum confined nanocrystals. The intrinsic phonon bottleneck of perovskite nanocrystals, that was previously reported by the authors (Nat Comms, 8, 2017), can be one of the causes favoring the appearance of MEG in such systems.*

I think that the results are well argued and are definitely valuable for the readership of Nature Communications. I therefore strongly recommend the acceptance of this work in Nature Communications

However, there are a few important aspects that should be addressed before publishing:

Response: We are delighted that reviewer shares our enthusiasm on the discoveries and greatly appreciate the reviewer's recommendation for publication of our work. The reviewer's critical comments are very helpful for improving our manuscript. We have carefully considered the reviewer's comments and would like to reply as follows:

Comment 1: *The authors always refer their results to multiples of the bandgap energy E_g . In the case of quantum confined nanocrystals, it would be more appropriate to refer to the exciton energy instead. If for simplicity reasons you want to keep the E_g nomenclature, you should at least comment on the exciton binding energy of these perovskite nanocrystals and, if this is nonnegligible, relate these results to multiples of the exciton energy.*

Response 1: We thank the reviewer for raising this point. In the pioneering reports on MEG studies, the E_g of NCs is defined by the 1s exciton absorption gap. We therefore used the E_g nomenclature to be consistent with these studies and for easy comparison by the readers to these works.

The value of E_g in our work is determined by the bleaching peak position of the exciton transition from transition absorption spectra, therefore, it corresponds to the excitonic absorption energy (i.e., exciton energy) and already includes the exciton binding energy (i.e., right third term of Eq.1 in SI). Furthermore, the calculated exciton binding energy is ~14-25 meV for our large to small FAPbI₃ NCs, is negligible as compared to exciton energies and therefore has a negligible effect (<1.5%) on the determination of MEG threshold in terms of E_g . Thus the

exciton energy can be used as the bandgap energy here. For clear understanding, we have therefore added “The small binding energies ($\sim 14\text{-}25$ meV for our large to small FAPbI₃ NCs) is negligible compared to the exciton energy. Here, we simply used the exciton energy (determined by TA spectra shown in Supplementary Fig. 3) as the bandgap energy.” in Note 1 of SI.

Comment 2: *Calculating the efficiency of the MEG process, the authors should clarify whether they refer to 2-exciton generation only or if they consider also possible higher order MEG (generating a number of excitons ≥ 3). These high-order processes should be taken into account, as they use as excitation energies also values that are higher than 3 times the bandgap energy.*

Response 2: We thank the reviewer for these constructive comments. We are not referring 2-exciton generation, higher order MEG is also considered. We have revised our previous unclear statement to the following sentence in the Supplementary Note 3 in SI: “For an ideal MEG process, β follows a staircase function, i.e., when the photon energy reaches the $N \times E_g$, maximum MEG yield is N (solid line in Supplementary Fig. 10a). For the non-ideal case (dashed lines in Supplementary Fig. 10a), as in our experiment, β increases linearly with $h\nu/E_g$, the slope represents the MEG slope efficiency (η_{MEG}).”

And recalculated Supplementary Fig. 10 based on the above method as shown below.

Supplementary Fig. 10 (a), multiple exciton generation yield as a function of $h\nu/E_g$ for an ideal case (max, solid line) and two non-ideal cases as examples (dashed lines). Calculated PCE under AM1.5 solar illumination as a function of E_g for MEG threshold of (a) $2E_g$ and (b) $3E_g$ at different MEG efficiencies.

Comment 3: *In the balance calculation reported in Fig. 3c, do you exclude the generation of 3 or more excitons (when allowed by the energy conservation)?*

Response 3: We have included the generation of 3 or more excitons, please refer to the response to comment 2.

Comment 4: *The authors should clarify in the text the method they have used to calculate the MEG QY.*

Response 4: Following the reviewer's suggestion, we have revised accordingly as shown below:

" R_{POP} is determined by the PB intensity ratio at the beginning of pump excitation to that at a delay time of 4 ns." in Fig. 2 caption is changed to:

" R_{POP} is determined by the PB intensity ratio at the beginning of pump excitation at a delay time of ~2 ps to that at a delay time of 4 ns and fitted with Eq. 1 (solid lines)."

and

"The R_{POP} ratios at the y-intercepts at above and below MEG threshold photon energy excitation thus facilitate the determination of MEG QY (Fig. 2d-f)." on page 7 of the main manuscript (paragraph 1, line 7) is changed to:

"MEG QY is determined by the ratio of the y-intercepts of fitting curves of R_{POP} (using Eq. 1) with $\langle N_0 \rangle \ll 0.1$ at above and below MEG threshold (Fig. 2d-f)."

Reviewer #2:

General Comments: *In the present manuscript, the authors explore the implications of their recent work (M.J. Li et al., Nat. Comm. 8, 14350 (2017)) showing a phonon bottleneck in the cooling of hot carriers in lead halide perovskite nanocrystals (NCs). The slowing of hot carrier cooling observed in their earlier work suggests that the efficiency of carrier multiplication (CM) should increase in lead halide perovskite nanocrystals compared to the bulk material. The observation of slower cooling with increasing confinement in their earlier report opens the possibility that the efficiency of CM could be higher in lead halide perovskites than in traditional inorganic NCs like PbS and PbSe. The present manuscript illustrates that these expectations are born out. Specifically, the threshold for the onset of CM in FAPbI₃ NCs is found to be as low as 2.25 ± 0.1 times the gap in 7.5 nm diameter NCs, while the slope efficiencies are observed to be as high as 75% (i.e., an average of 0.75 extra electron-hole pairs are generated per additional gap of photon energy above the CM threshold). These numbers are substantially better than in the most widely studied systems, PbS and PbSe, which had once been thought to offer the greatest potential for CM. Importantly, the efficiencies reported here are most favorable in the smallest dots, which suggests room for further improvement. This is again in contrast to the case of traditional inorganic NCs in which carrier cooling rates increase with increasing confinement, thereby negating the gains in Coulomb interactions and relaxation of momentum conservation that come with increased confinement and that would otherwise be expected to lead to increased CM efficiencies.*

The study of CM has struggled somewhat since the initial reports of high efficiencies in inorganic NCs were found to be erroneous. CM has been overtaken as a topic of study by singlet fission for a variety of reasons including perceptions of potential efficiencies and the fact that there have been clearer paths to design of molecules for singlet fission. There will be interest in the present work in boosting prospects for CM in a relatively simple material system. As such, this is a valuable contribution to the field. It should also be noted that the data in the present manuscript is very thorough, and great care appears to have been taken to obtain clean data.

Response: We are delighted that reviewer shares our enthusiasm on the discoveries. The reviewer's critical comments are very helpful for improving our manuscript. We have carefully considered the reviewer's comments and would like to reply as follows:

Comments 1: *It should be noted that Klimov's group has shown a CM threshold below 2.2 eV in PbSe/CdSe core/shell NCs (C.M. Cirloganu et al., Nat. Comm. 5, 4148 (2014)), which partly undercuts one of the main claims here, namely that the CM threshold is lower than in traditional inorganic NCs. However, the slope efficiencies were still only 25%. At least as important as the numbers is the fact that such core/shell NCs are fundamentally more complicated than the perovskite NCs studied here.*

Response 1: We thank the reviewer for pointing out this reference that we were not aware of. There are many reports on determining MEG thresholds of semiconductor nanostructures from

many different groups using different methods. Here we only compare with the classic semiconductor nanocrystals (i.e., PbS and PbSe NCs), which have been thoroughly investigated using the transient absorption method. The MEG thresholds were found to be at $\sim 3E_g$, and are consistent with our control experiments. We therefore refer them as “*benchmark materials in the MEG NCs discussion*”. Apart from the core-shell nature, we note that the method to determine the MEG threshold in this paper (Nat. Comm. 5, 4148 (2014)) was based on the TRPL approach. Nonetheless, we have added this reference in Table S2 of SI as an example of core/shell NCs.

We would also like to point out that our claim is on “surpassing the highest MEG efficiencies in strongly confined incumbents”, and not the MEG threshold. We are referring to the slope efficiency.

We have therefore cautioned our claim “surpassing the highest MEG efficiencies in strongly confined incumbents” and revised it to “surpassing the highest MEG **slope** efficiencies in strongly confined incumbents” in manuscript.

Comment 2: *The one general shortcoming I see in the paper is the absence of a quantitative explanation for the observations. That the CM efficiencies in FAPbI₃ NCs are greater than in the bulk was to be expected given the authors’ previous paper on the phonon bottleneck in lead halide perovskite nanocrystals. In lines 28-30, the authors write that “these NCs solve the conundrum over the contrasting requirements of enhancing Coulombic coupling and the density of states in strongly confined NCs.” The results don’t really solve the conundrum; they just avoid it by taking advantage of slowed cooling.*

Response 2: We thank the reviewer’s valuable comment. To make our statement clearer to readers, we have changed “these NCs solve the conundrum over the contrasting requirements of enhancing Coulombic coupling and the density of states in strongly confined NCs.” into “These perovskite NCs **circumvent the conundrum over the increased carrier cooling rates and reduced density of states which reversed the gains of enhanced Coulomb coupling and relaxation of momentum conservation in strongly confined conventional NCs.**” on page 3-4 of the revised manuscript. The comment on “quantitative explanation for the observations” is addressed in the next comment.

Comment 3: *Like many papers on CM, the paper offers more in the way of materials characterization than new scientific insights. A particularly glaring deficiency is the absence of discussion of the effective masses of electrons and holes. In the conventional picture, for a system characterized by single conduction and valence bands and in the case that the effective masses of electrons and holes are equal, the threshold for CM is $3E_g$. In other lead halide perovskites, the effective masses of electrons and holes are similar to each other (see, for*

example, Ref. 27 of the present manuscript and L.M. Herz, ACS Energy Letters 2, 1539 (2017)). As noted in lines 152-154 of the present manuscript, the authors of Ref. 27 did not observe MEG in CsPbI₃ under excitation at a photon energy of 2.65 E_g. In fact, it was in terms of the similar electron and hole masses that the failure to observe MEG in Ref. 27 was understood. From this perspective, the low CM threshold reported here is unexpected. It may be that the conventional picture is not relevant here, but such a statement would need to be justified.

Response 3: We greatly appreciate the reviewer's constructive comments.

In the conventional picture, MEG is believed to be more favourable for systems where the ratio of the electron/hole (or hole/electron) effective mass is large. This is based on the simple assumption that the lighter particle can take most of the excess excitation energy, when $m_e \gg m_h$ or $m_h \gg m_e$, the MEG threshold would be close to 2E_g.

However, there are also several detailed theoretical calculations (based on atomistic semiempirical pseudopotential method) predicting that systems with small and similar electron/hole effective masses are more favourable for slow hot-carrier cooling and therefore more efficient MEG (e.g., *J. Phys. Chem. Lett.* 2013, 4, 317; *Chem. Phys. Lett.* 2010, 496, 227).

As highlighted in Prof Eran Rabani's computational and theoretical work (*J. Phys. Chem. Lett.* 2013, 4, 317): "*It turns out that the intuitive assumption that the lighter particle takes most of the excitation energy is, in fact, incorrect. Indeed, transitions where the lighter particle takes the excess energy are much stronger than other transitions. However, the density of singly excited states, where the heavy particle takes the excess energy, is much larger. Thus, the effective oscillator strength of such transitions is larger, often by 2 orders of magnitude. Because the average Coulomb coupling of the heavier particle is significantly lower, the overall efficiency decreases when the two masses differ, consistent with the results shown in Figure 1.*"

Another recent computational work (*J. Phys. Chem. Lett.* 2017, 8, 3032) also predicted the low-threshold (close to 2E_g) MEG in MAPbI₃ NCs and the stronger Coulomb coupling between the initial single-exciton and final-biexciton states and longer hot-carrier cooling of highly excited states is more favourable for MEG. The calculated MEG process is on the time scale of tens fs, which agrees well with our measured <90 fs.

Supplementary Fig. 13 (a), Optimized cubic structure of FAPbI₃. (b), DFT calculated band structure of FAPbI₃.

We have now included DFT calculations (Supplementary Fig. 13). The electron and hole effective mass were calculated to be $0.14m_0$ and $0.20 m_0$, respectively. The small and similar effective mass of the electron and hole in the FAPbI₃ are thus favourable for slower hot-carrier cooling and more efficient MEG, which is in agreement with the theoretical predictions.

Based on the reviewer's suggestion, we have added the following discussion on the mechanism of efficient and low-threshold MEG in our perovskite NCs: "From our density function theory (DFT) calculations (Supplementary Fig. 13), the electron and hole effective masses are found to be $0.14m_0$ and $0.20m_0$, respectively. The small and similar effective mass of the electron and hole are predicted to be favourable for slow hot-carrier cooling and more efficient MEG.^{36, 37} Furthermore, another recent computational work³⁸ also demonstrated low-threshold MEG (close to $2E_g$) in methylammonium lead iodide (MAPbI₃) NCs. The authors attributed that the stronger Coulomb coupling between the initial single-exciton and final-biexciton states, and the longer hot-carrier cooling of highly excited states were more conducive for MEG. Their calculated MEG process was on the time scale of tens of fs, which agrees well with our measured value of <90 fs. Although, PbSe or PbS also have similar small carrier effective masses, the high MEG threshold can be explained by their faster hot-carrier cooling (Fig. 4c) and/or reduced density of final states in strongly confined NCs. We therefore ascribe the origins of the low threshold and efficient MEG in our intermediate-confined perovskite NCs to arise from the small and similar carrier effective masses, slow hot-carrier cooling and appropriate quantum confinement effect." on page 11 of manuscript and Supplementary Fig. 13 in the revised SI.

Comment 4: *The rest of my comments are about specific details of the paper.*

Panels c and d in Supplementary Figs. 5 – 7 are somewhat confusing. In particular, what is the physical meaning of using a linear fit for the A data, i.e., for? With the exception of the 7.5 nm NCs excited below the MEG threshold, the fits are clearly statistically inconsistent with the data

insofar as few of the error bars cross the fit line. Moreover, the inconsistencies are in different directions for different data sets. In Figs. 5c and 6d the A data seem roughly linear; in Fig. 5b the data appears to have a sub-linear dependence on fluence; and in Figs. 6c, 7c, and especially 7d, the A data appear to have a super-linear dependence on fluence.

Response 4: We thank the reviewer for his/her valuable comment.

The linear fit of A is only guide for eyes showing a possible trend that may exist in the figures. We acknowledge that the initial PB signal (i.e., A) only grows almost-linearly at low pump fluence. At high pump fluence, it will saturate. We agree with the reviewer that our linear fit for A maybe misleading to the readers, we have therefore deleted these fits.

Our MEG yield is determined at extremely low pump fluence (i.e, the y intercepts when $\langle N_0 \rangle \ll 0.1$), the inconsistent trends of A at high pump fluence would not affect the determination of MEG yield by A/B. Actually, the A/B data have already been provided in Panels e of Fig. 5-7. Therefore, all the different trends of A at high pump fluence ($\langle N_0 \rangle > 1$ to 2) will not affect the determination of MEG QY.

The following are the possible explanations for these different trends of A for different sized NCs.

The state-filling-induced transient absorption change can be expressed as:

$$-\Delta A = \sum_i (n_i^e + n_i^h) \alpha_{0,i}(\hbar\omega)$$

Where $\alpha_{0,i}(\hbar\omega)$ is the contribution of transition $\hbar\omega_i$ to the ground-state absorption at the probe wavelength, $n_i^e(n_i^h)$ are occupation numbers of the electron (hole) states involved in the transition. In the case of the twofold degeneracy in NCs, ideally, the sum of average populations of the lowest electron and hole states as a function of **initial** average e population can be presented as:

$$\langle n_{1s}^e + n_{em}^h \rangle = 2 \left[1 - e^{-\langle N \rangle} \left(1 + \frac{\langle N \rangle}{2} \right) \right]$$

Where, $\langle N \rangle$ is the average number of generated excitons per NC, which equals to $\langle N_0 \rangle \text{MEG}_{\text{QY}}$, $\langle N_0 \rangle$ is the average absorbed photons per NC. Therefore, at larger $\langle N \rangle$ at higher pump fluence or $\text{MEG}_{\text{QY}} > 1$, PB is easier to saturate (see Figure below), which is consistent with our observations of $\langle N_0 \rangle$ dependence of A (Fig. 5d) for smallest NCs with highest MEG_{QY} .

Figure R1. Initial TA amplitude as a function of pump fluence $\langle N_0 \rangle$.

It should also be noted that the above red and blue curves are calculated based on the constant MEG_{QY} and neglect the multiple exciton interactions at higher carrier densities. However, at higher pump fluence, the MEG_{QY} may be changed (may become larger due to stronger carrier interactions at higher carrier density), therefore the experimental may deviate from the pure state-filling model.

In addition, the fast surface trapping (reported on the time scale of tens to hundreds of fs) usually exists in NCs following photoexcitation. Therefore, there is also a competing process between hot-carrier cooling, MEG and surface trapping. As discussed in our previous paper (*Nat. Comms.*, 8, 14350, 2017), the slower hot-carrier cooling or the presence of the phonon bottleneck could be due to fewer surface traps. Nonetheless, details of competing process between hot-carrier cooling and surface trapping in perovskite NCs are still not clear, which will be investigated in the future. The superlinear increase of PB signal at higher pump fluence (Fig. 7c-d) for larger NCs with weak MEG effect could be due to the saturation of surface traps.

Therefore, these analysis suggest that the carrier density-dependent competing channels bestows a sublinear or superlinear behaviour of the initial peak PB signal with pump intensity, resulting in different trends for different sized NCs under different photon energies. The detailed investigation of these deviations at high pump fluence is beyond the scope of this work.

Comment 5: *The authors should state that the FA in FAPbI₃ refers to formamidinium the first time FAPbI₃ is mentioned in the manuscript, not just in the methods section.*

Response 5: We have changed “FAPbI₃ NCs” to “formamidinium lead iodide (FAPbI₃) NCs” in the introduction of the manuscript.

Comment 6: *I presume that the solid red and blue fits to RPOP are fits of Eq. 1. This is implied but not stated explicitly.*

Response 6: This is correct. To make our statement clearer to readers, we have revised accordingly as shown below:

“ R_{POP} is determined by the PB intensity ratio at the beginning of pump excitation to that at a delay time of 4 ns.” in Fig. 2 caption is changed to:

“ R_{POP} is determined by the PB intensity ratio at the beginning of pump excitation **at a delay time of ~2 ps to that at a delay time of 4 ns and fitted with Eq.1 (solid lines).**”

and

“The R_{POP} ratios at the y-intercepts at above and below MEG threshold photon energy excitation thus facilitates the determination of MEG QY (Fig. 2d-f).” in page 7 of manuscript is changed to:

“MEG QY is determined by the ratio of the y-intercepts of fitting curves of R_{POP} (using Eq.1) at $\langle N_0 \rangle \ll 0.1$ at above and below MEG threshold (Fig. 2d-f).”

Reviewer #3:

General Comments: *The manuscript "Low threshold and Efficient Multiple Exciton Generation in Halide Perovskite Nanocrystals" describes multiple exciton generation in colloidal FAPbI₃ nanocrystals. The claimed multiple exciton generation (MEG) threshold is very low ($2.25E_g$), being potentially relevant for photovoltaic applications. The positive effect on power conversion efficiency of solar cells is demonstrated by calculations for different MEG thresholds, showing that only the threshold of $2E_g$ is relevant.*

The efficiency of the MEG is relatively high due to the phonon bottleneck and slow cooling of hot carriers. The latter has been experimentally studied by the femtosecond transient absorption (TA) measurements, where the band edge bleaching shows a risetime of hundreds of femtoseconds. The dynamics of multiple excitons is studied by TA spectroscopy as well, where the band edge bleaching at pumping photon energies above $2E_g$ exhibits a fast component at the time scale of 100 picoseconds.

The conclusions are supported by experiments in a convincing way. The experiment description is detailed enough for reproduction.

Response: We are delighted that the reviewer shares our enthusiasm of the discoveries. We thank the reviewer for his/her time in reviewing our work and providing valuable feedback.

REVIEWERS' COMMENTS:

Reviewer #1 (Remarks to the Author):

The authors have adequately replied to my original comments and provided satisfactory answers to the points raised. I suggest acceptance of this work in its current form.

Reviewer #2 (Remarks to the Author):

From my perspective, the authors have thoroughly addressed the concerns raised by me (Reviewer #2) and Reviewer #1. As I noted previously, the results will be of interest to the broader community of researchers in photovoltaic applications of perovskites and those studying multiple exciton generation and related phenomena, and the data is thorough and of high quality. With their revisions, the authors have removed the less well founded or confusing elements of their analysis and arguments while grounding their arguments more solidly from a theoretical perspective.

- All reviewer comments are displayed in *dark orange italics*.
- All our responses are displayed in black.

Reviewer #1:

Reviewer #1 (Remarks to the Author):

The authors have adequately replied to my original comments and provided satisfactory answers to the points raised. I suggest acceptance of this work in its current form.

Response: We thank the reviewer for his time to review and help us strengthen our manuscript. We are delighted that he shares our enthusiasm about the findings in our work and we would like to thank reviewer's recommendation to publish our work.

Reviewer #2:

From my perspective, the authors have thoroughly addressed the concerns raised by me (Reviewer #2) and Reviewer #1. As I noted previously, the results will be of interest to the broader community of researchers in photovoltaic applications of perovskites and those studying multiple exciton generation and related phenomena, and the data is thorough and of high quality. With their revisions, the authors have removed the less well founded or confusing elements of their analysis and arguments while grounding their arguments more solidly from a theoretical perspective.

Response: We thank the reviewer for his time to review and help us strengthen our manuscript. We are delighted that he shares our enthusiasm about the findings in our work and we would like to thank reviewer's recommendation to publish our work.